# Optimization of Electropolishing Process Using Taguchi Robust Design for UNS N08367 in a Mixed Solution of Sulfuric Acid and Phosphoric Acid

**Hyun-Kyu Hwang [1] and Seong-Jong Kim [2],***

1 Graduate School, Mokpo National Maritime University, 91, Haeyangdaehak-ro, Mokpo-si 58628, Republic of Korea
2 Division of Marine Engineering, Mokpo National Maritime University, 91, Haeyangdaehak-ro, Mokpo-si 58628, Republic of Korea
* Correspondence: ksj@mmu.ac.kr

**Abstract:** The purpose of this investigation was to determine the optimal conditions for UNS N08367 electropolishing using the Taguchi method. The investigated factors were the electrolyte composition ratio, applied current density, and electrolyte temperature. Each factor was tested at three levels. Electropolishing was optimized using analysis of variance (ANOVA), signal-to-noise ratio (the smaller the better the characteristics), and surface analysis. The ANOVA results showed that among the three factors, only the electrolyte composition ratio was effective in surface planarization. The optimal conditions for electropolishing determined according to the signal-to-noise ratio were a sulfuric acid-to-phosphoric acid ratio of 2:8, a current density of 400 mA/cm$^2$, and an electrolyte temperature of 75 °C.

**Keywords:** UNS N08367; electropolishing; Taguchi method; ANOVA; signal-to-noise ratio





## 1. Introduction

With the growth of industries, such as the semiconductor, biomedical, food hygiene, and fine chemical device industries, the demand for special-purpose products (e.g., ultra-precision micro-parts, medical devices inserted into the human body, and microsensors) is increasing [1]. These products require advanced finishing technology to achieve surface quality that cannot be achieved with conventional mechanical polishing [2]. Mechanical polishing provides low surface roughness but cannot be used for products that require ultraclean and ultraprecise surfaces because it leaves impurities, fine grinding traces, and processing alteration layers on the surface of the workpiece [3]. Therefore, it is necessary to replace the finishing processes in which the tool and the workpiece are in direct contact with noncontact electropolishing methods based on electrochemical processing. In the 1960s, super-austenitic stainless steel was developed by increasing the chromium, molybdenum, and nitrogen contents, which provided a higher pitting resistance.

However, in the early stages of its development, super-austenitic stainless steel was not used due to its high manufacturing cost [4]. Recent industrial advancements have reduced the manufacturing cost of super-austenitic stainless steel, which is increasingly replacing general austenitic stainless steel. Accordingly, many electrochemical investigations into super austenitic stainless steel have been conducted [5–7], but there has been no research on electropolishing. Nevertheless, as micro-parts and high-purity gas containers used in corrosive environments require high corrosion resistance, and ultraclean and ultraprecise surfaces, it is necessary to optimize the process of super-austenitic stainless steel electropolishing. Since the electrochemical method was used for electropolishing, many parameters, such as electrolyte composition ratio, current density, temperature, polishing time, anode and cathode area, and stirring speed, were considered. In addition, the surface properties of the metal material vary depending on the electrolyte and component

content. In addition, since the surface is damaged when electropolishing is performed under excessive conditions, many experiments are required to select the appropriate conditions. However, since it is very difficult to conduct an experiment that considers numerous factors, a statistical analysis is applied. Analyzing data using a statistical method is very advantageous in terms of cost savings because information on quality improvement can be obtained with the minimum number of experiments, and it is a very useful method for optimizing electropolishing. Accordingly, other researchers performed statistical analyses on electropolishing [8–10].

Mahardika et al. investigated the optimal electropolishing conditions for titanium based on the applied potential, the content of ethanol, and the gap between the anode and the electrode [8]. Brent et al. explored the optimal electropolishing conditions for UNS S31603 using the Taguchi method with processing time, electrolyte composition, and temperature as factors [9]. Rokosz et al. electropolished duplex stainless steel and analyzed its surface using X-ray photoelectron spectroscopy [10]. However, although several studies have explored the optimal electropolishing conditions for various metals using statistical analyses, no studies have investigated the optimal conditions for super-austenitic stainless steel electropolishing using both statistical analysis (the Taguchi method) and surface analysis.

The purpose of this experiment was to analyze the effectiveness of each factor on the response value through various statistical analyses using the Taguchi robust design, as well as to calculate the optimal electropolishing conditions for super austenitic stainless steel using the signal-to-noise ratio. If the Taguchi method is used with many factors and levels, it can generate noise in an ANOVA. Therefore, it generally involves three to four factors and three levels [8,9,11], this research was designed with three factors and three levels. Statistical analysis methods for electropolishing involved an analysis of variance (ANOVA) using signal-to-noise ratio and the idea of the "smaller-the-better". The electropolished surface was also analyzed using a 3D microscope and a scanning electron microscope (SEM), and the findings were compared with the statistical analysis results.

## 2. Experimental Method

For UNS N08367 (AL-6XN), electropolishing was performed by applying the Taguchi method (Taguchi robust design) for the design of experiment with various conditions (factors, levels). The Taguchi method uses statistical terms for analysis, based on statistics. In this research, the parameters (electrolyte, current density, temperature) are defined as factors, and the variables of each factor are defined as levels.

Table 1 shows the chemical composition of the UNS N08367 specimen (AL-6XN) used in this experiment. The pitting resistance of stainless steel was evaluated using the pitting resistance equivalent index (PREN), and PREN was calculated using Equation (1) [12]:

$$PREN = \%Cr + 3.3\%Mo + 16\%N \tag{1}$$

The pitting resistance equivalent index of the specimen was 45.6—that is, about twice that of general austenitic stainless steel UNS S31603 (23.6). To process the specimen, thermal deformation was minimized using a fine-cutting machine supplied with cooling water. Following the processing of an exposed area of 1 cm$^2$, the specimen was mounted with an epoxy resin and polished using emery paper #220. Foreign substances generated after polishing were removed using acetone and ultrasonically washed in distilled water for 3 min. The specimen was dried in a dryer for 24 h and then used as a working electrode for electropolishing. The counter electrode was also made of UNS N08367. Since the effect of electropolishing varies according to the area ratio of the working and counter electrodes, the area ratio used conformed to the ASTM B912-02 standard [13]. An Ag/AgCl (saturated 3.3 M KCl) electrode was used as the reference electrode, and electropolishing was performed by applying a constant current.

The Taguchi method is a robust design for noise (environmental) factors that cannot be controlled through controllable factors, and is used to reduce surface roughness (improving quality). In addition, the Taguchi method is a quality improvement technology with the optimal factor and level using the signal-to-noise ratio and loss function [14].

The Taguchi design was created using an orthogonal arrangement table, and the experiments were performed accordingly. Based on the research on stainless steel by other researchers, this investigation selected the electrolyte composition ratio, current density, and temperature as the polishing conditions [15–18]. Each factor was selected at 3 levels. The three levels of the sulfuric acid–to–phosphoric acid ratio were 2:8, 3:7, and 4:6. The three applied current density levels were 200, 300, and 400 mA/cm$^2$. The three temperature levels were 70 °C, 75 °C, and 80 °C. The details are shown in Table 2.

The remaining conditions referred to are ASTM standards: the gap between the working electrode and the counter electrode was 5 mm, and the processing time was 5 min [13]. Since the surface properties of a metal material vary depending on component content, ASTM standards are for reference material only and did not necessarily perform as it is.

The loss function of the Taguchi method was applied to the principle of "the smaller-the-better". The ANOVA of the signal-to-noise ratio was performed using Minitab$^®$ 21 software. Surface analysis after electropolishing was performed using a 3D analytical microscope and SEM.

**Table 1.** Chemical compositions of UNS N08367 (wt%).

| Ni | Cr | Mo | C | Si | Mn | P | S | Cu | N | Fe |
|---|---|---|---|---|---|---|---|---|---|---|
| 24.62 | 20.6 | 6.44 | 0.015 | 0.27 | 0.72 | 0.017 | 0.001 | 0.53 | 0.232 | Bal. |

**Table 2.** Designed control factors and levels of UNS N08367.

| Factors | Unit | Level | | |
|---|---|---|---|---|
| Electrolyte (A) 95 wt% $H_2SO_4$ : 85 wt% $H_3PO_4$ | - | 1 (2:8) | 2 (3:7) | 3 (4:6) |
| Current density (B) | mA/cm$^2$ | 1 (200) | 2 (300) | 3 (400) |
| Temperature (C) | °C | 1 (70) | 2 (75) | 3 (80) |

## 3. Experimental Results and Discussion

Table 3 presents the orthogonal array used in this investigation and presents the electropolishing conditions. The orthogonal array method, which reduces the number of experiments required by making the number of each experimental condition (factor and level) the same, was used to detect factors affecting surface planarization and factors exerting interaction effects [19]. The control factors and levels were placed in the inner array, and uncontrollable factors were placed in the external array. The results (Ra, Rb, Rc) were recorded three times to minimize errors caused by environmental factors [20]. Moreover, each result value (Rx) was recorded as an intermediate value after measuring the surface roughness of three parts to further minimize errors. The factors and levels affecting surface planarization, resulting in different roughness values, were analyzed using ANOVA.

In the Taguchi method, a loss function is used to achieve the target quality, which can take the form of "the nominal-the-best", "the smaller-the-better", and "the larger-the-better" [21].

**Table 3.** Designed Taguchi orthogonal array of UNS N08367.

| | Inner Array | | | Outer Array: Roughness, μm | | |
| :---: | :---: | :---: | :---: | :---: | :---: | :---: |
| **Row** | **Electrolyte (H$_2$SO$_4$, 95 wt%)** | **Current Density (mA/cm$^2$)** | **Temp. (°C)** | | | |
| | **A** | **B** | **C** | **R1** | **R2** | **R3** |
| 1 | 1 | 1 | 1 | R$_1$ | R$_2$ | R$_3$ |
| 2 | 1 | 2 | 2 | R$_4$ | R$_5$ | R$_6$ |
| 3 | 1 | 3 | 3 | R$_7$ | R$_8$ | R$_9$ |
| 4 | 2 | 1 | 1 | R$_{10}$ | R$_{11}$ | R$_{12}$ |
| 5 | 2 | 2 | 2 | R$_{13}$ | R$_{14}$ | R$_{15}$ |
| 6 | 2 | 3 | 3 | R$_{16}$ | R$_{17}$ | R$_{18}$ |
| 7 | 3 | 1 | 1 | R$_{19}$ | R$_{20}$ | R$_{21}$ |
| 8 | 3 | 2 | 2 | R$_{22}$ | R$_{23}$ | R$_{24}$ |
| 9 | 3 | 3 | 3 | R$_{25}$ | R$_{26}$ | R$_{27}$ |

Table 4 presents the results of the surface roughness measurements after electropolishing and the signal-to-noise ratios using the principle that the "smaller-the- better". Following the electropolishing, the surface roughness was expressed in values ranging from 0.169 to 0.880 μm. The signal-to-noise ratio is used to determine the ratio of the force that interferes with the quality improvement during electropolishing [22]. The principle of "the smaller-the-better" was used because the specimen quality in this case depended on surface roughness after electropolishing. When the force in the experimental condition is stronger than the force that interferes with the quality improvement, a high reliability is obtained; thus, surface roughness is minimized when the signal-to-noise ratio is high. In terms of the "smaller-the-better", the further the surface roughness value (quality characteristic value) is from 0 (target value), the poorer the quality, and the loss function {L(y)} is the same as in Equation (2) [23]:

$$L(y) = k(y - 0)^2 \text{ (k : quality loss coefficient)} \tag{2}$$

**Table 4.** UNS N08367 roughness signal-to-noise ratios.

| | Inner Array | | | Outer Array: Roughness, μm | | | |
| :---: | :---: | :---: | :---: | :---: | :---: | :---: | :---: |
| **Row** | **Electrolyte (H$_2$SO$_4$, 95 wt%)** | **Current Density (mA/cm$^2$)** | **Temp. (°C)** | | | | **SN Ratios** |
| | **A** | **B** | **C** | **R1** | **R2** | **R3** | |
| 1 | 1 | 1 | 1 | 0.230 | 0.246 | 0.232 | 12.53 |
| 2 | 1 | 2 | 2 | 0.210 | 0.201 | 0.196 | 13.87 |
| 3 | 1 | 3 | 3 | 0.169 | 0.175 | 0.173 | 15.27 |
| 4 | 2 | 1 | 1 | 0.232 | 0.239 | 0.237 | 12.54 |
| 5 | 2 | 2 | 2 | 0.500 | 0.483 | 0.474 | 6.27 |
| 6 | 2 | 3 | 3 | 0.228 | 0.232 | 0.224 | 12.84 |
| 7 | 3 | 1 | 1 | 0.872 | 0.857 | 0.880 | 1.21 |
| 8 | 3 | 2 | 2 | 0.527 | 0.510 | 0.547 | 5.54 |
| 9 | 3 | 3 | 3 | 0.569 | 0.549 | 0.537 | 5.16 |

In Equation (2), the loss function {L(y)} is the surface roughness (y) value after electropolishing under one condition [24]. The idea of the smaller the better reflects the fact that

if each loss function {L(y$_i$)} is small, the quality is good, and that the overall surface roughness value(y$_i$) is also small. Equation (3) reflects the substitution of the surface roughness value after electropolishing under each condition:

$$
\begin{aligned}
L(y_1) &= k(y_1 - 0)^2 \\
L(y_2) &= k(y_2 - 0)^2 \\
&\vdots \\
L(y_i) &= k(y_i - 0)^2
\end{aligned}
\tag{3}
$$

In Equation (4), the mean square deviation (MSD) is the average value after electropolishing under all conditions. Therefore, Equation (4) is the average of Equation (3) and is calculated as follows:

$$
\begin{aligned}
MSD &= k(y_1 - 0)^2 \\
&+ k(y_2 - 0)^2 \\
&\vdots \\
&+ k(y_i - 0)^2 \\
&= k\left[\tfrac{1}{n}\sum_{i=1}^{n}(y_i - 0)^2\right]
\end{aligned}
\tag{4}
$$

Since 10 log (MSD) can be used as the signal-to-noise ratio, the quality loss minimization (smaller-the-better) is determined by Equation (5), which is the expression for maximizing the signal-to-noise ratio [24]:

$$
S/NRatio = -10\log\left(\frac{1}{n}\sum_{i=1}^{n} y_i^2\right)
\tag{5}
$$

The signal-to-noise ratio was the highest with a sulfuric acid–to–phosphoric acid ratio of 2:8 (Level 1), an applied current density of 400 mA/cm$^2$ (Level 3), and an electrolyte temperature of 80 °C (Level 3). This condition resulted in the lowest quality loss after electropolishing, with a surface roughness value closest to the target value of 0.

The results of the signal-to-noise ratio using ANOVA performed to analyze and compare the contributions of the control factors affecting surface planarization during electropolishing are shown in Table 5. ANOVA is important for the interpretation of the F-value (Fisher–Snedecor distribution) and the P-value (probability). These values are calculated using the degrees of freedom, adjusted sum of squares, and adjusted mean sum of squares. Specifically, ANOVA determines the contribution (variability) of each factor to surface flattening. Variability is calculated as the deviation and is the difference between the average value of the response value (surface roughness) and the response value. The variation is calculated as the sum of squaring the difference between the surface roughness $y$ obtained under each condition and the average value $\bar{\bar{y}}$ obtained from all conditions (sum of squares), which can be calculated using Equation (6) [25]:

$$
SS = \sum_{i=1}^{a}\sum_{j=1}^{n_i}\left(y_{ij} - \bar{\bar{y}}\right)^2
\tag{6}
$$

Generally, the total sum of squares is calculated by adding the sum of squares of each factor, interaction, and error. Here, the interaction is that two or more different factors combine to affect the response value (surface roughness). In this research, the interactions are between the electrolyte composition ratio and applied current density (A*B), applied current density and temperature (B*C), temperature and electrolyte composition ratio (C*A), and electrolyte composition ratio, applied current density and temperature (A*B*C). Interactions in the Taguchi method were ignored because they were less important than other factors [6,26]. Therefore, the total sum of squares was calculated, as in Equation (7) [6]:

$$
SS_T = SS_A + SS_B + SS_C + SS_E
\tag{7}
$$

**Table 5.** ANOVA results for the roughness signal-to-noise ratio. ($R^2$ = 84.2).

| Source | DF | Adj SS | Adj MS | F-Value | *p*-Value |
|---|---|---|---|---|---|
| Linear | 3 | 0.38191 | 0.12730 | 8.93 | 0.019 |
| A | 1 | 0.30781 | 0.30781 | 21.59 | 0.006 |
| B | 1 | 0.02257 | 0.02257 | 1.58 | 0.264 |
| C | 1 | 0.05152 | 0.05152 | 3.61 | 0.116 |
| Error | 5 | 0.07130 | 0.01426 | | |
| Total | 8 | 0.45320 | | | |

The sum of squares is adjusted and calculated regardless of the calculation order. The adjusted mean sum of squares is calculated by dividing the adjusted sum of squares by the degrees of freedom (the amount of information in the data used to estimate statistics) [27]. The F-value is calculated by dividing the adjusted mean sum of squares of each factor by the adjusted mean sum of the square of the error [28]. The F-value is used as a numerical value to determine whether there is a correlation between surface roughness and each factor. When the F-value is high, the fluctuations in surface planarization are large, which means that the correlation is high. The correlation order of the three factors is considered to be A > C > B. The *p*-value is the result of standardizing the F-value on a graph that follows a 95% normal distribution and is used to test the null and alternative hypotheses [29]. The *p*-value is an important numerical value for the analysis of the effectiveness of each factor for the response value. No electropolishing parameter can be prioritized because all parameters work in a complex way, and the conditions and roughness values in Table 4 cannot be used to analyze the quantitative effectiveness of each factor. In addition, since there has been no research on the electropolishing of super austenitic stainless steel, quantitative values for effective factor selection will be useful to other researchers. The null hypothesis in this experiment was that each factor would affect surface flattening, and the alternative hypothesis was the opposite. A *p*-value of less than 0.05 would mean that the factor affected surface roughness. Thus, the alternative hypothesis would be rejected. In this experiment, only the *p*-value of factor A (electrolyte composition ratio) was less than 0.05; therefore, only this factor had a significant effect on surface planarization. The *p*-values of factors B and C were 0.264 and 0.116, respectively, indicating that these factors had weaker effects on surface planarization than factor A. Nevertheless, they had considerable effects at approximately 74% and 88% confidence levels, respectively.

Figure 1 depicts the results of a Pareto chart analysis with a 95% confidence level of the normal distribution for the effective factor. Pareto charts are commonly used to prioritize and act on major issues. In this research, the Pareto chart was used to set priorities for the factors affecting surface planarization. The advantage of a Pareto chart is that since the numerical values are standardized by dividing the effect size of each factor by the scatter, comparisons can be made regardless of the unit [30]. As a result of setting the effective level on surface planarization to 0.05 in the Pareto chart, the threshold value was calculated as 2.1, with factors above this threshold considered to affect surface planarization. A Pareto chart also reflects a form of statistical analysis that supports the evaluation of the validity of the *p*-values. The results of the analysis presented that the effectiveness of each factor for the response value was the same as the *p*-value, similarly, the applied current density had the lowest effect on the surface planarization. When current density was applied, planarization proceeded, as the protrusions on the rough metal surface were removed. As the applied current density increased, the protrusions were removed rapidly, and the long-term effect of the applied current density was insignificant; therefore, it did not contribute significantly to surface planarization [31].

Table 6 is a response table for the signal-to-noise ratio ("the smaller-the-better") of surface roughness created to derive the optimal process conditions for super-austenitic stainless steel electropolishing. The delta value was calculated as the difference between the maximum and minimum values among the levels of each factor, with the highest value set at 1, followed by 2 and 3. The delta value of factor A was 9.921, indicating the strongest effect on surface planarization, whereas that of factor B was the lowest (2.529). The trend was identical to the Pareto chart. In general, the effect of temperature on the corrosion reaction is expressed by Equation (8) (Arrhenius equation) [32]:

$$K = A\exp[-(E_a/RT)] \tag{8}$$

{K: rate constant, A: pre-exponential factor, $E_a$: apparent activation energy, R: molar gas constant, T: thermodynamic absolute temperature (in Kelvin or degree Rankine)}.

According to this equation, as the temperature increases, the value of the rate constant K increases, and the electrochemical reaction is initiated. The rise in temperature also increases the number of collisions between molecules, so the molecules above the activation energy (the minimum energy required for the dissolution reaction) increase, promoting the ionization reaction [33]. As the temperature increases, the surface becomes flatter, but the electrical resistance is sufficiently low to allow current flow in the electrolyte above a certain temperature. It is thought that the dissolution action, which preferentially reacts with the metal protrusions, does not increase significantly with sufficient current flow. Therefore, the electropolishing effect at 80 °C was weaker than at 75 °C.

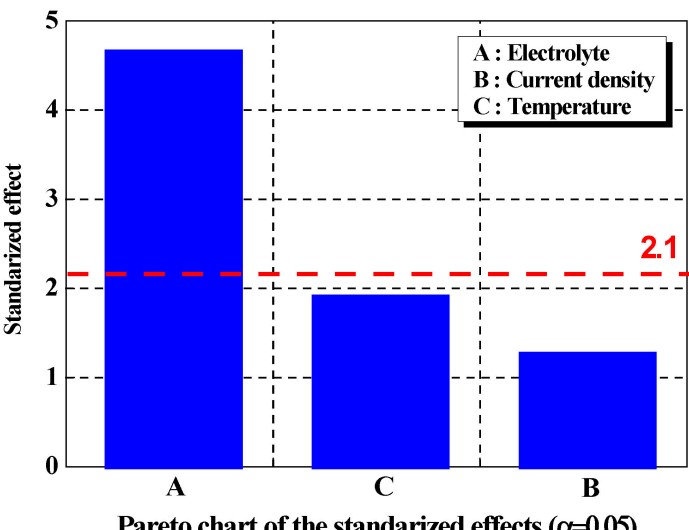

**Figure 1.** Pareto chart of the standardized effects of factors with signal-to-noise ratios.

**Table 6.** Response table of roughness signal-to-noise ratio—smaller the better.

| Level | Electrolyte | Current Density | Temperature |
|:---:|:---:|:---:|:---:|
| 1 | 13.895 | 8.764 | 10.307 |
| 2 | 10.551 | 8.563 | 10.527 |
| 3 | 3.973 | 11.092 | 7.585 |
| Delta | 9.921 | 2.529 | 2.942 |
| Rank | 1 | 3 | 2 |

Figure 2 presents the main effect plot of the signal-to-noise ratio. The main effect plot is the result of applying the principle of "the smaller-the-better," meaning that the surface is flatter with higher signal-to-noise ratios. As the concentration of sulfuric acid in the electrolyte increased, the signal-to-noise ratio decreased, so the degree of surface

planarization decreased. In the case of applied current density, the signal-to-noise ratio at 300 mA/cm² was slightly lower than at 200 mA/cm², so the degree of surface flattening slightly decreased, while its value at 400 mA/cm² was higher than at 300 mA/cm², so the surface was smooth. In the case of temperature, the signal-to-noise ratio at 75 °C was slightly higher than at 70 °C, so surface planarization slightly increased. Conversely, its value at 80 °C was significantly lower than at 75 °C, and the surface became rather rough. If the temperature is higher than necessary, the surface is damaged due to excessive ionization. Therefore, since the optimal condition for electropolishing is the condition that combines the highest signal-to-noise ratios obtained from all factors, the electrolyte composition ratio of 2:8, the current density of 400 mA/cm², and the electrolyte temperature of 75 °C yielded the flattest surface.

Table 7 indicates a comparison between the signal-to-noise ratio calculated using Minitab® 21 for the predicted optimal condition shown in Figure 2 and the experimentally calculated signal-to-noise ratio. The predicted optimal condition was determined by combining the highest signal-to-noise ratios obtained from all factors. Since the predicted value (16.58) was higher than the experimental value (15.27), it was considered the optimal electropolishing condition. Therefore, the optimal factor levels were the electrolyte composition ratio of 2:8, the current density of 400 mA/cm², and the electrolyte temperature of 75 °C.

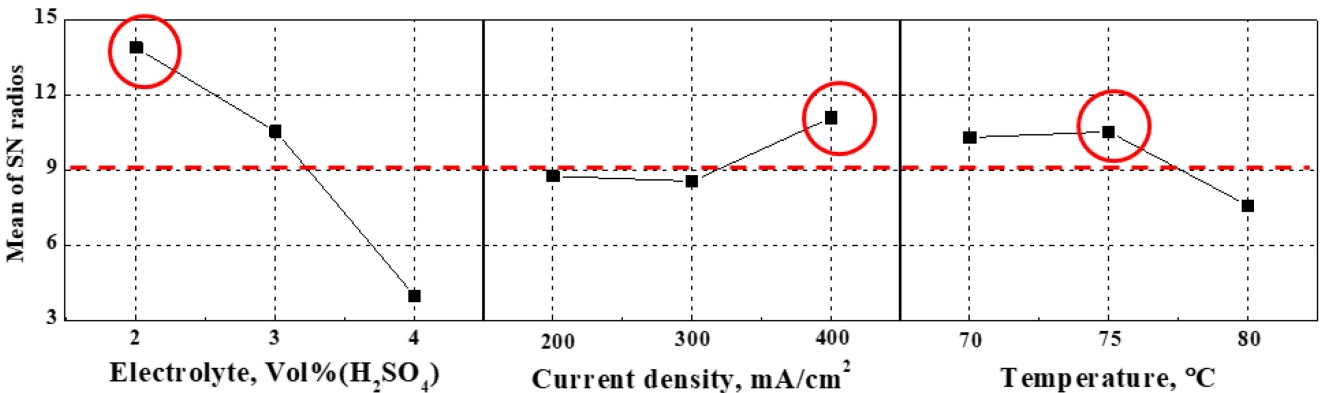

**Figure 2.** Main effects plot of the mean signal-to-noise ratios with surface roughness.

**Table 7.** Optimization conditions with roughness signal-to-noise ratios.

|  | Electrolyte | Current Density | Temperature | SN Ratios |
|---|---|---|---|---|
| Experimental value | 2 | 400 | 80 | 15.27 |
| Predicted value | 2 | 400 | 75 | 16.58 |

Figure 3 indicates the potential over time for each electropolishing condition. The electropolishing conditions were labeled to simplify the references to them. In the case of a single condition, the three levels of the electrolyte composition ratio (factor A) were denoted as $A_3$. When electropolishing was performed under two or more conditions, it was expressed as $A_1B_3C_2$ (electrolyte composition ratio: one level; current density [factor B]: three levels; temperature [factor C]: two levels). Electropolishing should be performed under conditions in which the potential value remains stable over time. Accordingly, the point at which the potential changes rapidly, is indicated by a blue circle. In the $A_1$ condition, the voltage fluctuated during the first 1 min but remained stable thereafter. In the $A_2$ condition, the potential value in all conditions except for $A_2B_2C_3$ stabilized after a sudden change between 3 and 4 min. In the $A_3$ condition, the potential value was stable over time in all conditions except for $A_3B_3C_2$ during the first 1.5 min. Therefore, stable values were maintained in all electropolishing conditions after 4 min, indicating that the electrochemical reaction proceeded stably. Stainless steel containing Mo and Cr forms a

bipolar passive oxide film [34]. The two-phase membrane consists of a cation-selective layer ($CrO_4^{2-}$, $MoO_4^{2-}$) and an anion-selective layer. Since super-austenitic stainless steel contains large amounts of Mo and Cr to form a cation-selective layer, the passivation film is more stable.

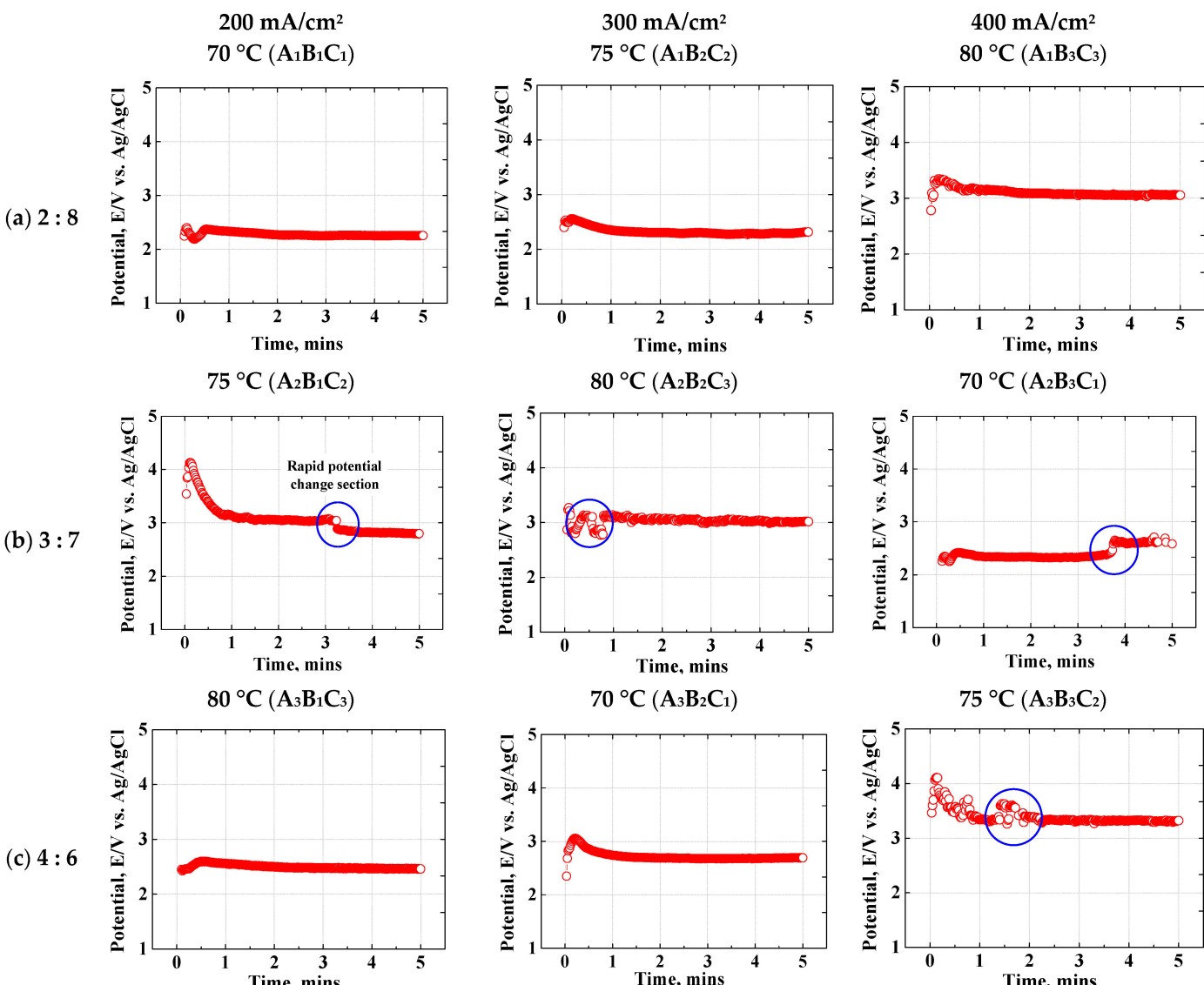

**Figure 3.** Galvanostatic electropolishing experiment using an electrolyte containing sulfuric acid (95 wt%) and phosphoric acid (85 wt%) in ratios of (**a**) 2:8, (**b**) 3:7, and (**c**) 4:6.

Figure 4 presents the results of the electropolished surface observations using SEM. In all conditions except $A_3B_1C_3$ and $A_3C_3B_2$, the surface was smooth and very clean, with no mechanical polishing marks. In the $A_3B_1C_3$ and $A_3C_3B_2$ conditions, the metal surface was damaged (red rectangular area) due to excessive polishing. In general, corrosion of stainless steel grows with pitting and intergranular corrosion. Pits grow in parts where a potential difference is generated by impurities on the metal surface. However, since the electropolished surface in this experiment was clean and the electrolyte component was free from halogen elements that cause pitting (e.g., Cl, F, and Br), it was damaged in a relatively poor part.

Figure 5 exhibits the metal surface profiles and the height differences between mountains and valleys in each electropolishing condition. The height difference was calculated as the difference between the highest and lowest values in each profile. In the $A_1B_1C_1$ condition, the surface was relatively rough, with mountains and valleys. It can be assumed that in this condition, electropolishing was not performed sufficiently because the sulfuric acid content, current density, and temperature were low. Conversely, in the $A_1B_3C_3$ condition, electropolishing was sufficient due to the higher current density and temperature, and the surface (0.76 μm) was flatter than in the $A_1B_1C_1$ condition (2.56 μm). In the $A_2$ condition, which had a higher proportion of sulfuric acid, damage to the flat surface caused by local corrosion was observed. In the $A_3$ condition, which had the highest sulfuric acid content, the metal surface was overall rough and irregularly damaged. Consequently, the profile and height difference analysis showed that $A_1B_3C_3$ was the optimal electropolishing condition, producing the flattest surface. Moreover, the largest height difference ($A_3B_1C_3$, 6.16 μm) was about eight times greater than the smallest difference ($A_1B_3C_3$, 0.76 μm).

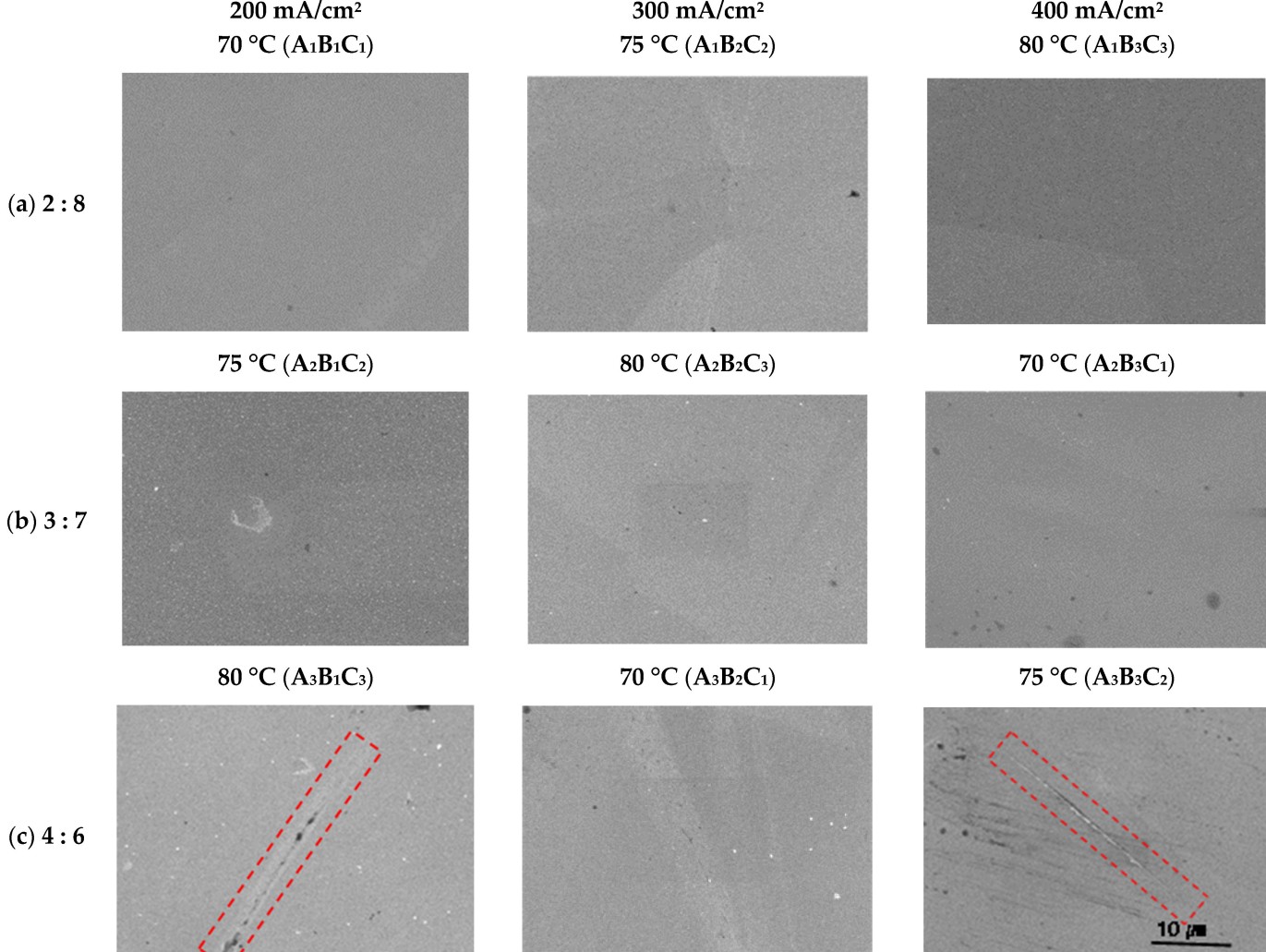

**Figure 4.** Surface morphologies after electropolishing in an electrolyte containing sulfuric acid (95 wt%) and phosphoric acid (85 wt%). $A_3B_1C_3$, $A_3B_3C_2$: damaged, the rest: smooth and cleanliness.

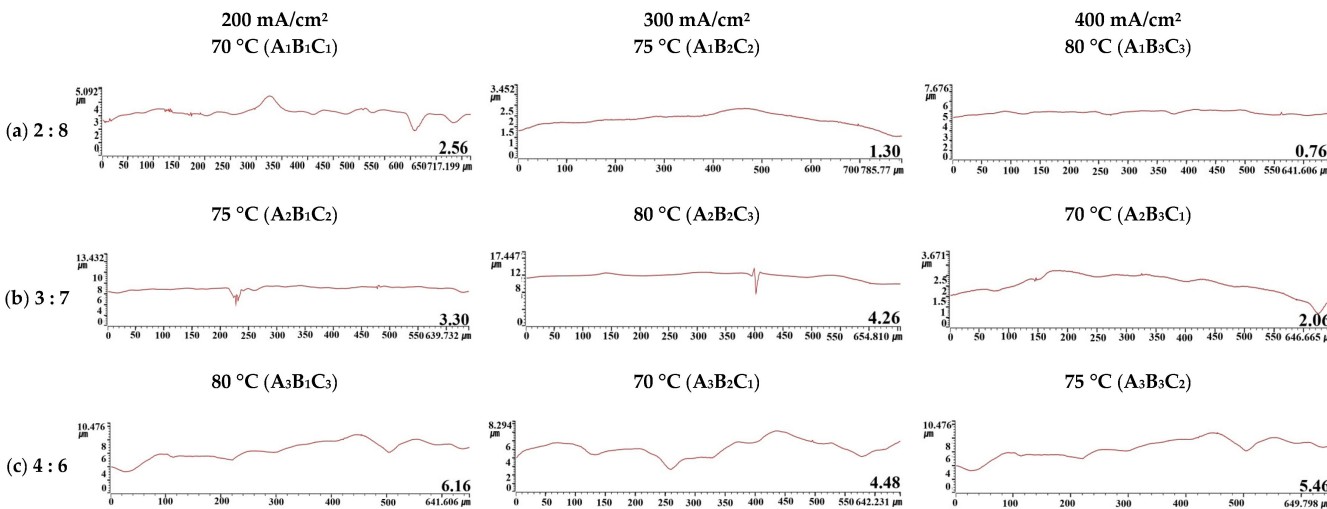

**Figure 5.** Height differences between mountains and valleys after electropolishing in an electrolyte containing sulfuric acid (95 wt%) and phosphoric acid (85 wt%), Unit: μm. $A_1B_3C_3$: the flattest, $A_3B_1C_3$: the wildest.

Figure 6 presents the surface roughness and a depth histogram after electropolishing. A depth histogram is a graph showing the distribution (y-axis) from the shallowest to the greatest depth (x-axis). It can be used to infer the state of a metal surface from the mean and dispersion, so is useful when calculating the optimal conditions for electropolishing [35]. Since the depth histogram in this experiment followed a normal distribution, information on surface planarization was obtained using the mean and dispersion. The mean—the most frequent value in the depth histogram—represented depth, and the dispersion indicated the degree to which the depth distribution was scattered from the average depth. The lower the mean value and dispersion, the flatter the surface. The depth histogram confirmed that $A_1B_3C_3$ was the optimal electropolishing condition. Conversely, the $A_3B_1C_3$ condition had a high mean value since it caused the most severe damage among all conditions. It also had the greatest dispersion because the overall damage was irregular.

To estimate the degree of surface roughness reduction after electropolishing, the surface roughness improvement rate was calculated using Equation (9) and Figure 7 [36]:

$$100 - \left( \frac{\text{Roughness after Electropoling}}{\text{Roughness before Electropoling}} \times 100 \right) = \text{Improvement rate of roughness, \%} \qquad (9)$$

The mechanical polishing surface roughness before electropolishing is 0.55 μm. $A_1B_3C_3$ was the condition with the greatest improvement in surface roughness (about 69.4%; 0.17 μm), whereas $A_3B_2C_1$ yielded the smallest improvement (about 4.5%; 0.53 μm). $A_3B_1C_3$ (0.87 μm) and $A_3B_3C_2$ (0.57 μm) had negative roughness improvement rates because roughness increased after electropolishing due to surface damage. In the $A_1$ condition, surface roughness decreased as the current density and temperature increased, whereas in the $A_2$ and $A_3$ conditions, it decreased in all conditions except $A_2B_3C_1$. It can be assumed that a clear improvement rate did not appear due to the interactions between the three factors.

In general, the selection criteria for the conditions of the electropolishing process were determined when the surface roughness improved by 50%, compared to mechanical polishing [37], and when the roughness was less than 0.8 μm, which is the standard in the semiconductor industry [38]. Therefore, the conditions selected in this investigation (roughness: 0.17 μm, surface roughness improvement rate: about 69%) were considered sufficient for the electropolishing process. However, the predicted signal-to-noise ratio for electropolishing was higher than the experimental condition, so it was considered the optimal polishing condition.

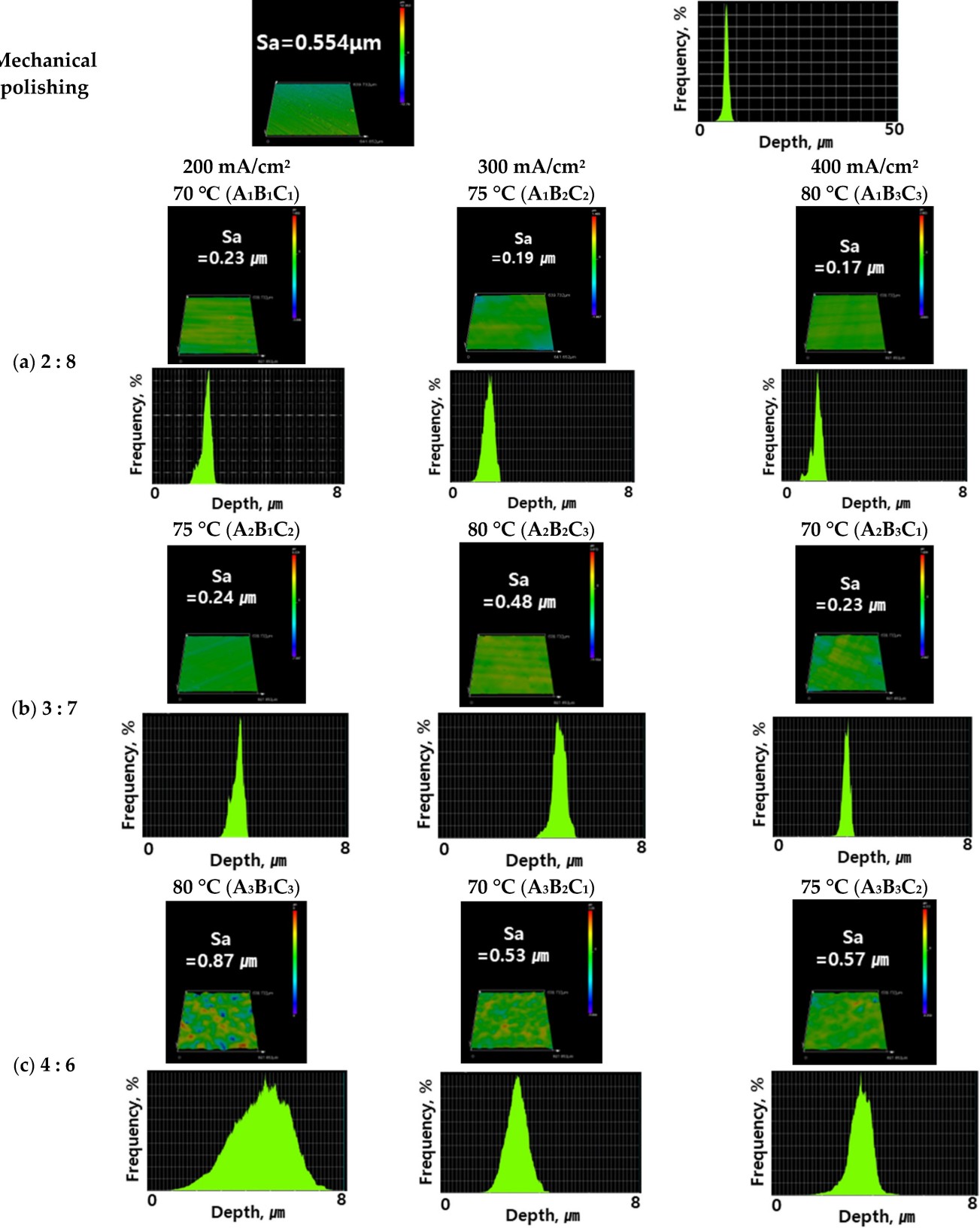

**Figure 6.** 3D analysis and depth histogram of damaged surfaces after electropolishing in an electrolyte containing sulfuric acid (95 wt%) and phosphoric acid (85 wt%). $A_1B_3C_3$: The mean and spread are the smallest, $A_3B_1C_3$: The mean and spread are the largest.

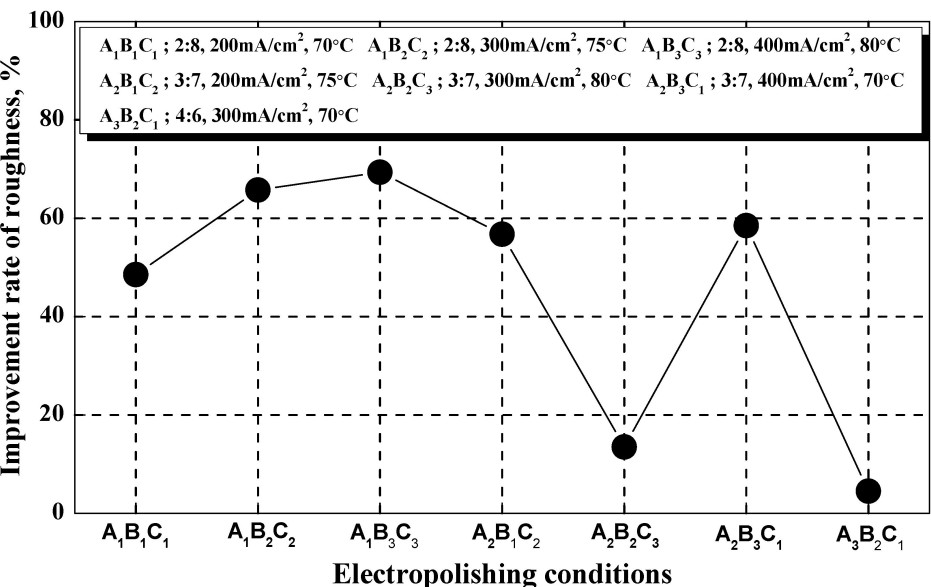

**Figure 7.** Comparison of surface roughness improvement rates after electropolishing.

## 4. Conclusions

As a result of variance analysis using the Taguchi robustness design in investigating the electropolishing conditions for super austenitic stainless steel, only the electrolyte component ratio was effective in surface planarization, as can be seen from the *p*-value and delta value of the signal-to-noise ratio. The order of effectiveness was electrolyte component ratio > electrolyte temperature > current density. The microscopic analysis presented that the electrolyte composition ratio was the most important factor. When the ratio of sulfuric acid and phosphoric acid was adjusted, the electropolishing conditions with the largest roughness and the smallest roughness were determined, confirming the ANOVA results.

Electropolishing using a phosphoric acid-to-sulfuric acid ratio of 2:8, an applied current density of 400 mA/cm$^2$, and a temperature of 80 °C ($A_1B_2C_2$) yielded a surface improvement of more than 69% over mechanical polishing and can therefore be considered a suitable condition.

However, the expected signal-to-noise ratio of electropolishing is higher than that of electropolishing in this experimental condition, so a phosphoric acid-to-sulfuric acid ratio of 2:8, an applied current density of 400 mA/cm$^2$, and a temperature of 75 °C ($A_1B_2C_2$) can be considered to constitute the optimal condition.

**Author Contributions:** Conceptualization, S.-J.K. and H.-K.H.; methodology, S.-J.K. and H.-K.H.; validation, S.-J.K. and H.-K.H.; investigation, S.-J.K. and H.-K.H.; resources, H.-K.H.; data curation, S.-J.K.; writing—original draft preparation, H.-K.H.; writing—review and editing, S.-J.K.; visualization, H.-K.H.; supervision, S.-J.K. All authors have read and agreed to the published version of the manuscript.

**Funding:** This research was funded by a research project titled 'Demonstration of aftertreatment systems of Ship's air pollutant (NO$_x$/SO$_x$/PM) and establishment of their certification system' from the Ministry of Oceans and Fisheries, Korea grant number [20190402]. The APC was funded by a research project titled 'Demonstration of aftertreatment systems of Ship's air pollutant (NO$_x$/SO$_x$/PM) and establishment of their certification system' from the Ministry of Oceans and Fisheries, Korea.

**Institutional Review Board Statement:** Not applicable.

**Informed Consent Statement:** Not applicable.

**Data Availability Statement:** The data are not publicly available due to the project requirements.

**Conflicts of Interest:** The authors declare no conflict of interest.

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
