# Peer review of "Optimization of Electropolishing Process Using Taguchi Robust Design for UNS N08367 in a Mixed Solution of Sulfuric Acid and Phosphoric Acid"

_coatings, doi:10.3390/coatings13020312_

Round 1
Reviewer 1 Report
The authors have investigated the optimization condition of electrolyte composition ratio, applied current density, and electrolyte temperature for UNS N08367 electropolishing using Taguchi method. The authors concluded that the sulfuric acid–to–phosphoric acid ratio of 2:8, a current density of 400 mA/cm2, and an electrolyte temperature of 75°C is the optimal electropolishing condition. The work has good merit; however, it needs improvements.
(1) The same information has been written repetitively in a few places, which needs the manuscript to be polished.
(2) In Table 2, please write what levels 1, 2, and 3 represent in the same way you have written for factors A, B, and C.
(3) Please explain how Eq. (2) is related to Eq. (1).
(4) On page 5 of 13, line 152, the authors say that the interaction is ignored. What interaction are the authors talking about?
(5) The labels are hard to read in many figures. Please fix it. (6) It is hard to make the conclusion that the authors have because only three ratios, three temperatures, and three current densities are taken into consideration. For example, if 2:8 looks promising, then please have a few more ratios, such as 1:9, 3:17, and 5:15 as well for further confirmation. My numbers are just a few examples for reference.
(7) Equation 7 is not readable. Please fix it.
Author Response
Please see the attachment(PDF file)
1. Responses
2. Manuscript
Thank you for your review.

Reviewer 2 Report
In this manuscript, the authors investigated the optimum electropolishing parameters for stainless steel using both statistical analysis and surface analysis. The presentation of manuscript is clear. This study may be of interest to the readers of Coatings. In my opinion, the article has several deficiencies, and it needs some corrections before being considered for publication.
- In introduction part, please mention the importance of the statistical analysis.
- Some texts are not read clearly in Figure 2, 3 and 5. Please revise them.
- There are typos. Please check the language and writing quality of paper.
For example:
“Daguchi” in keywords
Author Response

(The authors gave the same response as above.)

Reviewer 3 Report
The manuscript investigates the effect of some parameters on the standard electropolishing process, using Taguchi Robust Design. The application of statistical analysis is not justified in this work, because the analysis is performed on only 9 input data. The parameters providing the optimal electropolishing outcome can be easily deduced from the measured surface roughness data (Table 3), by using common sense. The scientific contribution of the manuscript is very low, because it discusses the parameter conditions already defined in the appropriate ASTM standard. So, there is no improvement or development of the electropolishing process. Some comments are:
Page 2, line 68: explain the calculation of the pitting resistance equivalent index.
Page 4, line 135: “The signal-to-noise ratio was highest with a sulfuric acid–to–phosphoric acid ratio of 2:8, an applied current density of 400 mA/cm2, and an electrolyte temperature of 80°C. This condition resulted in the lowest quality loss after electropolishing, with a surface roughness value closest to the target value of 0.” One can recognize the optimal polishing parameters simply by looking at the surface roughness data in Table 3. There is no need for “signal-to-noise ratio” tool in this work.
Page 5, line 169: “In this experiment, only the P-value of factor A (electrolyte composition ratio) was less than 0.05; therefore, only this factor had a significant effect on surface planarization. The P-values of factors B and C were 0.264 and 0.116, respectively, indicating that these factors had weaker effects on surface planarization than factor A.” One can make this conclusion simply by looking at Table 3, with 9 results. There is no need for statistical processing.
Page 6, line 180: “As a result of setting the effective level on surface planarization to 0.05 in the Pareto chart, the threshold value was calculated as 2.1, with factors above this threshold considered to affect surface planarization. The electrolyte composition ratio had the highest correlation with surface planarization, whereas the applied current density had the lowest.” One can make this conclusion simply by looking at Table 3, with 9 results. There is no need for statistical processing.
Page 7, line 213: “Therefore, the electropolishing effect at 80°C was weaker than at 75°C.” What is the scientific contribution of this conclusion? The temperature of 75°C is already defined in ASTM standard.
- ASTM B912 defines a typical electropolishing solution and operating conditions. The optimal electrolyte composition (acid ratio) recommended in this manuscript, differs from the one prescribed in the standard. Explain this discrepancy, bearing in mind that prior to preparing an ASTM standard, many experiments are performed to define the optimal process parameters.
Author Response

(The authors gave the same response as above.)

Round 2
Reviewer 1 Report
The authors have significantly improved the manuscript. The revised version looks a far better than the original submission. I am still not happy with Figure 5, however. The axes levels are not readable even with 150% magnification of the text. Please make appropriate modifications.
Author Response

(The authors gave the same response as above.)

Reviewer 3 Report
I believe the manuscript has been sufficiently improved to warrant publication in Coatings.Author Response
Please see the attachment(PDF file)
1. Responses
2. Manuscript
Thank you for your review.
